# Estimation of the Network Reliability for a Stochastic Cold Chain Network with Multi-State Travel Time

**Thi-Phuong Nguyen [1], Chin-Lung Huang [2] and Yi-Kuei Lin [2,3,4,5,\*]**

1    Master Program in Smart Manufacturing and Applied Information Science,
     National Chin-Yi University of Technology, Taichung 411070, Taiwan; phuongnt@gm.ncut.edu.tw
2    Department of Industrial Engineering and Management, National Yang Ming Chiao Tung University,
     Hsinchu 300093, Taiwan; zaxzax0148.mg10@nycu.edu.tw
3    Department of Business Administration, Asia University, Taichung 413305, Taiwan
4    Department of Medical Research, China Medical University Hospital, China Medical University,
     Taichung 404333, Taiwan
5    Department of Industrial Engineering and Management, Chaoyang University of Technology,
     Taichung 413310, Taiwan
\*    Correspondence: yklin@nycu.edu.tw

**Abstract:** A stochastic cold chain (SCC) is a common supply chain in real life that emphasizes the need for commodities to arrive fresh within time constraints. In previous research on supply chains, the time factor was regarded as a fixed number. However, the travel time is a stochastic factor due to traffic and weather conditions during the delivery. Therefore, this paper concentrates on the two multi-state factors simultaneously. Network reliability is one of the performance indexes used to assess the cold chain efficacy, defined as the probability that the flow of SCC can satisfy the demand within the delivery time threshold. The SCC with two multi-state factors is modeled as a stochastic cold chain network with multi-state travel time (SCCNMT). To calculate the network reliability of an SCCNMT, we will calculate the demand reliability and time reliability separately, treating them as independent events, and multiply the demand and time reliability to estimate the network reliability of the two multi-state factors.

**Keywords:** stochastic cold chains (SCC); network reliability; multi-state travel time; two multi-state factors

## 1. Introduction

As technological progress advances and generations change, the logistics industry has become an essential part of daily life [1]. In recent years, competition among businesses has intensified. To expand their customer base and improve profits, companies aim to cut costs, shorten delivery times, improve product quality, and offer personalized products. Therefore, the concept of the cold chain [2,3] is currently trending in the logistics industry. In the past, the reliability of the cold chain network was considered multi-state based on the number of carriers, without taking other factors into account. However, a stochastic cold chain network with multi-state travel time (SCCNMT) takes into account two stochastic factors simultaneously. This research does not consider time as a fixed constant, but instead views it as a multi-state factor.

A stochastic network is a real system, such as transportation systems, communication systems and supply chain systems [4–6]. In a cold chain network, we consider suppliers, logistics companies, and retail stores as nodes, and the path between each node as an arc. When the network demand is given, we can calculate the probability of successfully transmitting the flow from the source to the sink, which is the reliability of cold chain networks.

To calculate the reliability of cold chain networks [7–9], we only consider the multi-state flow caused by the occupied orders. However, time factors such as travel time, are affected by personnel and environmental factors. In the previous study [10,11], reliability was calculated without considering the multi-state travel time. Therefore, this research aims to develop a new reliability measure for a cold chain network that can simultaneously consider both the multi-state flow and travel time. The reliability of a cold chain network can be used as a performance indicator to assist cold chain businesses with order planning and decision-making.

Previous research on stochastic supply chain networks has explored various aspects of network reliability. Huang [8] studied the different states of available transportation on different roads and proposed a network performance evaluation algorithm for inventory issues. Lin et al. [7] investigated the use of different transportation modes in the network and considered the characteristic of goods being damaged. Niu et al. [12] further examined the cost concept by incorporating transportation and damage costs into the study and separately discussing the damage rates on different routes. Lin et al. [9] specifically discussed perishable goods, calculated the flow rate that meets the damage rate and ensured that the product can meet demand under the possibility of damage. As stochastic networks have been studied by previous studies with only a single multi-state factor, this study attempts to extend the research by evaluating the network reliability under two multi-state factors.

This study aims to evaluate the reliability of a cold chain throughout the entire production process. First, we model the entire cold chain as a network, with nodes representing cold chain suppliers, logistics companies, and retail stores, and each node connecting a pair of arcs. We consider the route for a product chain as a path and use the solution to the network problem to calculate the reliability of a cold chain network. In the past, this research addressed the issue of multi-state flow but did not account for the time factor as a multi-state variable [12–14]. Therefore, this study extends the approach to calculate the reliability of a cold chain network for two stochastic variables. The ultimate goal of this research is to determine the network reliability for actual products and account for both flow and time factors simultaneously.

The paper is divided into five sections, organized as follows: In this section, the introduction is presented. In Section 2, a stochastic cold chain network with a multi-state travel time (SCCNMT) model is developed. An algorithm is further proposed in Section 3 to evaluate the network reliability with multi-state flow and travel time. In Section 4, a simple numerical example of a practical cold chain is presented. Lastly, in Section 5, the remarks of the thesis are presented.

## 2. Model Construction for an SCCNMT

In this section, the notations and assumptions are introduced first.

### 2.1. Notations and Assumptions

Let $G \equiv (\mathbf{N}, \mathbf{A}, M)$ represent an SCCNMT with $\mathbf{N}$ being a set of perfect reliable nodes including suppliers, logistics companies, and retail stores, $\mathbf{A} = \{a_q \mid q = 1, 2, \ldots, z\}$ being a set of $z$ arcs, and $M = (M_q \mid q = 1, 2, \ldots, z)$ with $M_q$ being the maximal capacity of each arc. At each arc $a_q$, there are several identical carriers [7–9]. The capacity vector $X$ denotes the number of identical trucks in each arc, and $x_q$ takes a value from $\{0, 1, 2, \ldots, M_q\}$. A cold chain network is stochastic because each truck on the arc may be occupied by the other orders. To evaluate the network reliability of cold chain networks in an SCCNMT, the $G$ model developed in this research satisfies the following assumptions:

I.    The flow and time units of an SCCNMT are integer [15].
II.   No nodes, including suppliers, logistics companies and retail stores, provide inventory service.
III.  In such an SCCNMT, only one commodity is provided.

IV.    The capacity of any component is statistically independent [15].
V.    Flow on the network G satisfies the flow-conservation law [16].
VI.    The identical type of truck throughout the entire transportation process is used.

### 2.2. Total Delivery Time

A cold chain network involves several processes that a product must go through from supplies to retail stores. Firstly, the product is delivered to the logistics companies, where it undergoes the first travel time. Once it arrives at the logistics company, the product is unloaded and awaits loading until all goods have arrived for transportation. Afterward, the product undergoes the second travel time to reach the retail stores. The service time of the entire cold chain network is illustrated in Figure 1.

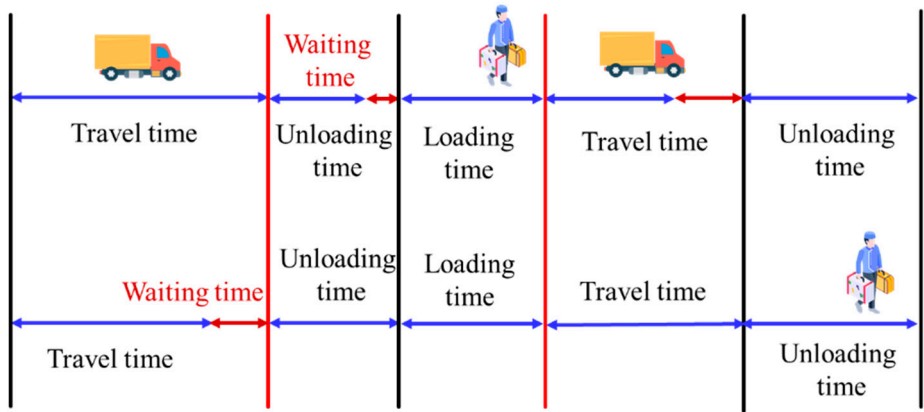

**Figure 1.** Service time concept in a cold chain.

The time required to deliver products to the retail stores can be divided into three parts. The first part is the travel time from suppliers to the logistics companies, taking into consideration the maximum travel time on the way, denoted by $\max\{v_b\}$. Here, *b* is a range from 1 to the arcs before the logistics companies s. After all goods are delivered, unloading and loading services begin. Hence, the second and third parts consider the unloading and loading times, and since all goods depart from the logistics company at the same time, the maximum time among all paths after logistics company is considered, denoted by $(U + L)$ and $\max\{v_q\}$.

### 2.3. The Capacity Vector

An SCCNMT exists in more than two suppliers and retail stores. The vector $X = (x_1, x_2, \ldots, x_q)$ is defined as the capacity vector of *G*, where $x_q$ represents the current used truck number of $a_q$. Here, $a_q$ is an integer random variable with a maximum value of $M_q$. An SCCNMT can be derived into two areas by logistic companies. One area focuses on supplying logistics companies with products to meet the demand of retail stores. The other area is for logistic companies to deliver products to retail stores. Due to the pressure of time, the capacity must consider both delivery time and demand simultaneously. If the flow vector *F* satisfies the following Equations (1) and (2), it is said to be feasible under $M_q$,

$$\left\lceil \sum_{o=1}^{w} \sum_{k=1}^{p} \left\{ f_o^k \middle| a_q \in P_o^k \right\} \right\rceil \leq M_q, \text{ for } q = 1, 2, \ldots, z, \tag{1}$$

$$\max\left\{f_o^k \big| a_q \in P_o^k\right\} \leq \left\lceil \frac{I-\Delta}{U} \right\rceil \text{for} o = 1, 2, \ldots, w; \ k = 1, 2, \ldots, p;$$
$$q = s+1, \ s+2, \ldots, z; \ b = 1, 2, \ldots, s, \text{ where } \Delta = \max\{v_b\} + (U+L) + \max\{v_q\}. \tag{2}$$

Equation (1) represents that the demand cannot exceed its maximal capacity $M_q$. Equation (2) is inverted by the total delivery time to ensure the number of trucks that can satisfy the flow on time.

Similarly, to be feasible under the capacity vector $X = (x_1, x_2, \ldots, x_q)$, the flow vector $F$ must satisfy the following Equations (3) and (4),

$$x_q = \left\lceil \sum_{o=1}^{w} \sum_{k=1}^{p} \left\{ f_o^k \big| a_q \in P_o^k \right\} \right\rceil, \text{ for } q = 1, 2, \ldots, s, \tag{3}$$

$$x_q = \left\lceil \frac{U*\max\left\{f_o^k \big| a_q \in P_o^k\right\}}{I-\Delta} \right\rceil \text{for} o = 1, 2, \ldots, w; \ k = 1, 2, \ldots, p;$$
$$q = s+1, \ s+2, \ldots, z; \ b = 1, 2, \ldots, s, \text{ where } \Delta = \max\{v_b\} + (U+L) + \max\{v_q\}. \tag{4}$$

### 2.4. Minimal Capacity Vectors and Demand Reliability

Let the set of capacity vectors that satisfy the demand vector $D = (d^1, d^2, \ldots, d^k)$ within time constraints be denoted as $G$. The demand reliability $R_D$ is defined as the probability that an SCCNMT can satisfy the demand under the maximum time constraint. Thus, $R_D$ can be represented using Formula (5) as follows,

$$R_D = \sum \Pr\{X | X \in \boldsymbol{\Omega}\}, \tag{5}$$

where $\Pr\{X\} = \Pr\{x_1\} \times \Pr\{x_2\} \times \ldots \times \Pr\{x_q\}$ with assumption 4.

However, when the network size is very large, it is inefficient to calculate network reliability by enumerating all $X$ in the set $\boldsymbol{\Omega}$ and summing their probabilities. Instead, we can identify the minimal capacity vector from those capacity vectors $X$ that satisfy the demand within the max time constraint and regard them as MCV. If there are $\gamma$ MCVs, the formulation of demand reliability can be revised. In order to obtain the MCVs, we need to go through the following steps and present in Table 1 how to obtain the process of $\boldsymbol{\Omega}_{\min}$. The demand reliability can be evaluated using the recursive sum of disjoint products (RSDP) algorithm [17,18]. Therefore, we can apply the RSDP algorithm to calculate demand reliability, as shown in Formula (6),

$$R_D = \Pr\left\{ \bigcup_{i=1}^{\gamma} \{X | X \geq X_i\} \right\}. \tag{6}$$

**Table 1.** The process of obtaining the $\boldsymbol{\Omega}_{\min}$.

| | |
|---|---|
| Line 1: | Suppose there are $\gamma$ delivery vector $X$ in $\boldsymbol{\Omega}$<br>Set $I = \boldsymbol{\Omega}_{\min} = \varnothing$ ($I$ is stack, which stores the MCV index) |
| Line 2: | For $i = 1$ to $\gamma$ with $i \notin I$ |
| Line 3: | For $j = i+1$ to $\gamma$ with $i \notin I$ |
| Line 4: | If $X_i \leq X_j$, $X_i$ is not a MCV and belongs to $\boldsymbol{\Omega}_{\min}$. |
| Line 5: | Else $X_i$ belongs to $\boldsymbol{\Omega}_{\min}$, $I \leftarrow I \cup \{i\}$ and go to Line 6. |
| Line 6: | End |
| Line 7: | $\boldsymbol{\Omega}_{\min} \leftarrow \boldsymbol{\Omega}_{\min} \cup X_i$ |
| Line 8: | End |

Various methods can be used to calculate the probability of a union set, including the inclusion–exclusion principle [17], state space decomposition method [18], sum of disjoint products [19], and recursive sum of disjoint products [17]. In this study, we employ the RSDP to obtain the reliability, which represents the probability of meeting time constraints.

### 2.5. Time Reliability

After considering the multi-state travel time, the travel time for each arc will follow its own probability distribution, unlike in previous research where it was treated as a constant value. Therefore, it is possible to calculate the probability of arriving at a specific destination within a certain time for a given transportation path based on the multi-state travel times for each arc. Considering all time factors, we can obtain the feasible travel time vector *V* using the following Equations (7) and (8),

$$v_q^{\min} \leq v_q \leq v_q^{\max} \text{ for } q = 1, 2, \ldots, z, \tag{7}$$

$$\left\{ v_b + v_c \big| a_b, a_c \in P_o^k \right\} \leq I^*, \text{ for } b = 1, 2, \ldots, s; c = s+1, s+2, \ldots, z. \tag{8}$$

Through constraint (7) and (8), we can derive travel time upper bound vector *S* in *V*. In order to avoid complex and inefficient calculations, similar to the last subsection, we also need to find a travel time upper bound as the same as MCVs. However, since time and capacity are different, we are seeking an upper bound solution, and thus, we need to use Table 2 to obtain the travel time upper bound vector. Each travel time upper bound vector represents a feasible configuration that can meet the current travel time. Therefore, if we obtain $\rho$ travel time upper bound vectors, we can use Equation (9) to calculate time reliability $R_T$, which represents that an SCCNMT can deliver within the time threshold; this can be developed as follows:

$$R_T = \Pr\left\{ \bigcup_{h=1}^{\rho} \{S|S \leq S_h\} \right\}. \tag{9}$$

**Table 2.** The process of obtaining the travel time upper bound vector.

| | |
|---|---|
| Line 1: | Suppose there are $\rho$ travel time vector *S* <br> Set $I = \mathbf{S} = \varnothing$ (I is stack, which stores the travel time upper bound vector index) |
| Line 2: | For $i = 1$ to $\rho$ with $i \notin I$ |
| Line 3: | For $j = i + 1$ to $\gamma$ with $i \notin I$ |
| Line 4: | If $V_i \geq V_j$, $V_i$ is not a travel time upper bound and belongs to **S**. |
| Line 5: | Else $V_i$ transform into vector $S_i$ and belongs to **S**, $I \leftarrow I \cup \{i\}$ and go to Line 6. |
| Line 6: | End |
| Line 7: | $\mathbf{S} \leftarrow \mathbf{S} \cup V_i$ |
| Line 8: | End |

### 2.6. Estimate the Network Reliability with Two Multi-State Factors

In the last subsection, we calculated the demand reliability $R_D$, which represents satisfying the demand within the maximum time constraints. We need to calculate the time reliability $R_T$ to consider the multi-state travel time and estimate the network reliability with the two factors. The time reliability $R_T$ [20–22] considers the time factors under a cold chain, so we need to understand every detail in a cold chain. A cold chain transportation of a product from the supplier to the retail stores requires passing through the travel time from the supplier to the logistics company and from the logistics company to the retail stores. In addition, during the transportation process, the product needs to be unloaded, picked, and handed over to the carrier again, and there may be a gap for each step. Finally,

we need to determine a threshold I to limit the latest arrival time. To estimate the network reliability under the two multi-state factors can be based on Formula (10),

$$R_{D,\,T} = R_D \times R_T \tag{10}$$

### 3. Algorithm

An algorithm to evaluate the network reliability for an SCCNMT model under time constraint *I* is proposed as follows (Algorithms 1–3):

| **Algorithm 1:** demand reliability $R_D$ |
| --- |

**Input:** $G \equiv (\mathbf{N}, \mathbf{A}, M)$, $D = (d^1, d^2, \ldots, d^p)$, $v_q$, $U$, $L$, $I$.

**Output:** demand reliability $R_D$

Step 0. Find all feasible flow vectors $F = (f_1^1, f_2^1, \ldots, f_w^1, f_1^2, f_2^2, \ldots, f_w^2, f_1^p, f_2^p, \ldots f_w^p)$　satisfying

$$\sum_{k=1}^{p}\sum_{o=1}^{w} f_o^k = d^k. \tag{11}$$

Step 1. Check whether the flow vectors *F* exceeds the maximal capacity $M_q$ of each arc or not.

(1.1) Check whether the maximal capacity of each arc has been exceeded by the number of vehicles used via

$$\left\lceil \sum_{o=1}^{w}\sum_{k=1}^{p}\{f_o^k \,|\, a_q \in P_o^k\} \right\rceil \le M_q, \text{ for } q = 1,\,2,\,\ldots,\,z. \tag{12}$$

(1.2) Check whether the truck departing from the logistics company can arrive within the given time via

$$\max\{f_o^k \,|\, a_q \in P_o^k\} \le \left\lceil \frac{I - \Delta}{U} \right\rceil, \text{ for } o = 1, 2, \ldots, w; k = 1, 2, \ldots, p;$$

$$q = s+1, s+2, \ldots, z; b = 1, 2, \ldots, s, \text{ where } \Delta = \max\{v_b\} + (U+L) + \max\{v_q\} \tag{13}$$

Step 2. Generate the capacity vectors $X = (x_1, x_2, \ldots, x_z)$ by following the procedure below.

(2.1) Transform each *F* into capacity vector *X* before the logistics company via

$$x_q = \left\lceil \sum_{o=1}^{w}\sum_{k=1}^{p}\{f_o^k \,|\, a_q \in P_o^k\} \right\rceil, \text{ for } q = 1,\,2,\,\ldots,\,s. \tag{14}$$

(2.2) Transform each *F* into capacity vector *X* after the logistics company while taking into consideration time constraints via

$$x_q = \left\lceil \frac{U * \max\{f_o^k \,|\, a_q \in P_o^k\}}{I - \Delta} \right\rceil, \text{ for } o = 1, 2, \ldots, w; k = 1, 2, \ldots, p;$$

$$q = s+1, s+2, \ldots, z; b = 1, 2, \ldots, s, \text{ where } \Delta = \max\{v_b\} + (U+L) + \max\{v_q\} \tag{15}$$

(2.3) Obtain capacity vectors *X* in **Ω** by steps 2.1 and 2.2, and eliminate other non-MCVs in **Ω** using vector operations.

Step 3. Assume there are *z* MCVs. The demand reliability $R_D$ in an SCCNMT using the RSDP can be obtained via

$$R_D = \Pr\left\{ \bigcup_{i=1}^{z} \left\{ X \,\middle|\, X \ge X_i \right\} \right\} \tag{16}$$

---

**Algorithm 2:** time reliability $R_T$

**Input:** $G \equiv (\mathbf{N}, \mathbf{A}, M)$, $D = (d^1, d^2, \ldots, d^p)$, $v_q$, $U$, $L$, $I$.

**Output:** time reliability $R_T$

Step 1. Calculate the currently allocatable travel time by considering the average loading and unloading time from the time threshold $I$ by Equation (17),

$$I^* = I - (U + 2 * L) \tag{17}$$

Step 2. Find the travel time upper bound vectors.

(2.1) Find the feasible travel time vectors $V$ for each route using Equations (18) and (19).

$$v_q^{\min} \leq v_q \leq v_q^{\max}, \text{ for } q = 1, 2, \ldots, z,$$

$$\{v_b + v_c \,|\, a_b, \, a_c \in P_o^k\} \leq I^*, \text{ for } b = 1, 2, \ldots, s; \, c = s+1, s+2, \ldots, z. \tag{18}$$

(2.2) Obtain travel time vectors $V$ by step 2.1 and using vector operations to eliminate other non-travel time upper bound vectors.

$$\max\{f_o^k \,|\, a_q \in P_o^k\} \leq \left\lceil \frac{I - \Delta}{U} \right\rceil, \text{ for } o = 1, 2, \ldots, w; \, k = 1, 2, \ldots, p;$$

$q = s + 1, s + 2, \ldots, z; b = 1, 2, \ldots, s,$ where $\Delta = \max\{v_b\} + (U + L) + \max\{v_q\}$ $\tag{19}$

Step 3. Assume there are $\rho$ travel time upper bound vectors. The time reliability $R_T$ in an SCCNMT using the RSDP can be obtained via

$$R_T = \Pr\left\{ \bigcup_{i=1}^{\rho} \{S \,|\, S \leq S_i\} \right\} \tag{20}$$

---

**Algorithm 3:** network reliability $R_{D, T}$

**Input:** demand reliability $R_D$, time reliability $R_T$

**Output:** network reliability $R_{D, T}$

$$R_{D, T} = R_D \times R_T \tag{21}$$

Figure 2 is provided to illustrate the entire algorithm.

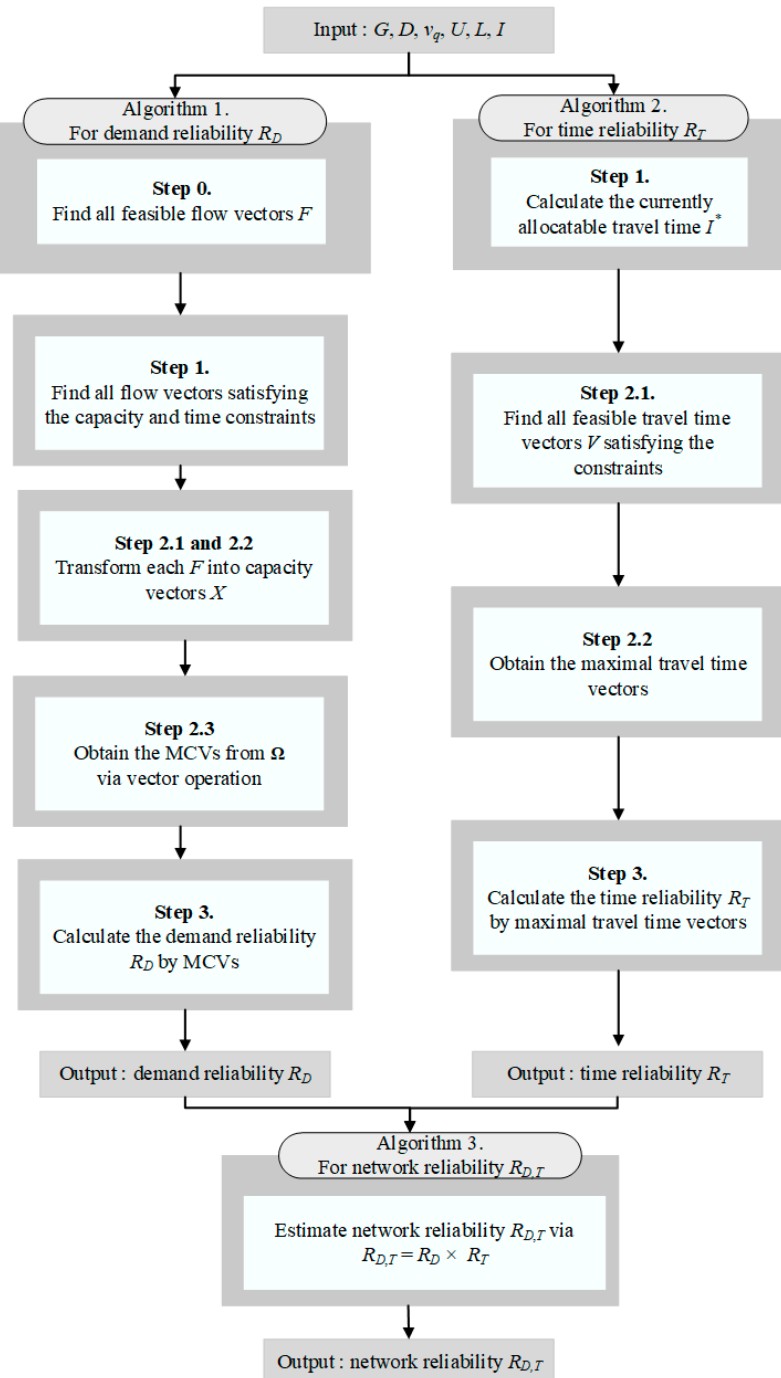

**Figure 2.** The procedure of the algorithm.

## 4. A Numerical Example

We use a simple numerical example to demonstrate an SCCNMT model, including a simple cold chain network. Figure 3 shows a simple cold chain network to illustrate an SCCNMT model, which consists of two suppliers, two logistics companies, three retail stores, and ten arcs. Table 3 shows all the time factors dealt with within the logistics enterprise. Tables 4 and 5 list the data on the capacity and arrival distribution of each arc.

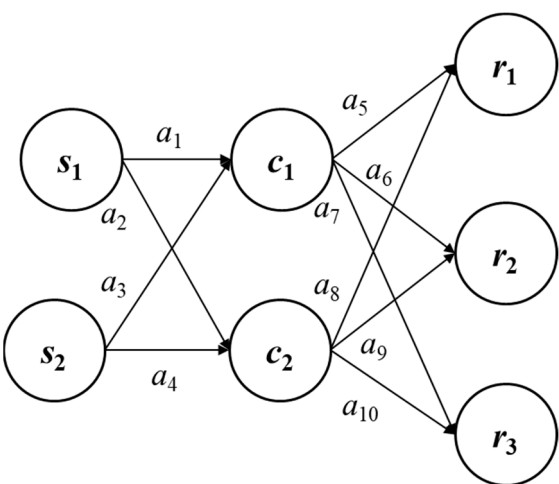

**Figure 3.** An SCCNMT example.

**Table 3.** Data of the service time in the logistics company.

| Time Factor | Average Time (mins) |
|---|---|
| Unloading | 15 |
| Loading | 20 |

**Table 4.** Truck data of the numerical example.

| Arc | Capacity | Probability |
|---|---|---|
| $a_1, a_2$ | 0 | 0.7 |
| $a_3, a_4$ | 1 | 0.1 |
| | 2 | 0.15 |
| | 3 | 0.05 |
| $a_5, a_6$ | 0 | 0.7 |
| $a_7, a_8$ | 1 | 0.2 |
| | 2 | 0.05 |
| | 3 | 0.05 |
| $a_9, a_{10}$ | 0 | 0.6 |
| | 1 | 0.2 |
| | 2 | 0.05 |
| | 3 | 0.15 |

**Table 5.** Travel time and arrival probability of each arc.

| Arc | Travel Time | Arrival Probability |
|---|---|---|
| $a_1, a_2$ | 35 | 0.7 |
| | 37 | 0.1 |
| | 40 | 0.1 |
| | 45 | 0.1 |
| $a_3, a_4$ | 27 | 0.8 |
| | 30 | 0.05 |
| | 32 | 0.05 |
| | 35 | 0.1 |
| $a_5, a_6$ | 35 | 0.9 |
| | 37 | 0.05 |
| | 40 | 0.025 |
| | 45 | 0.025 |

**Table 5.** *Cont.*

| Arc | Travel Time | Arrival Probability |
|---|---|---|
| $a_7, a_8$ | 20 | 0.8 |
| | 25 | 0.05 |
| | 27 | 0.1 |
| | 30 | 0.05 |
| $a_9, a_{10}$ | 30 | 0.9 |
| | 33 | 0.025 |
| | 35 | 0.05 |
| | 40 | 0.025 |

The following algorithm is used to obtain the network reliability of an SCCNMT given $D$ = (2, 2, 2), $I$ = 130 min, $M$ = (3, 3, 3, 3, 4, 4, 4, 3, 3, 3). There are 12 delivery paths in this network $P^1_1 = \{a_1, a_5\}$, $P^1_2 = \{a_3, a_5\}$, $P^1_3 = \{a_2, a_8\}$, $P^1_4 = \{a_4, a_8\}$, $P^2_1 = \{a_1, a_6\}$, $P^2_2 = \{a_3, a_6\}$, $P^2_3 = \{a_2, a_9\}$, $P^2_4 = \{a_4, a_9\}$, $P^3_1 = \{a_1, a_7\}$, $P^3_2 = \{a_3, a_7\}$, $P^3_3 = \{a_2, a_{10}\}$, $P^3_4 = \{a_4, a_{10}\}$ (Algorithms 4–6).

---

**Algorithm 4:** demand reliability $R$ (2, 2, 2)

---

Step 0. Find all feasible flow vectors $F$ = $(f^1_1, f^1_2, f^1_3, f^1_4, f^2_1, f^2_2, f^2_3, f^2_4, f^3_1, f^3_2, f^3_3, f^3_4)$ satisfying

$$f^1_1 + f^1_2 + f^1_3 + f^1_4 = 2,$$
$$f^2_1 + f^2_2 + f^2_3 + f^2_4 = 2, \quad (22)$$
$$f^3_1 + f^3_2 + f^3_3 + f^3_4 = 2.$$

This step generates 1000 flow vectors, which are shown in the first column of Table 6.

Step 1. Check whether the flow vectors $F$ exceeds the maximal capacity $M_q$ of each arc or not.

(1.1) Check whether the maximal capacity of each arc has been exceeded by the number of vehicles used via

$$\left\lceil f^1_1 + f^2_1 + f^3_1 \right\rceil \le 3,$$
$$\left\lceil f^1_3 + f^2_3 + f^3_3 \right\rceil \le 3,$$
$$\left\lceil f^1_2 + f^2_2 + f^3_2 \right\rceil \le 3,$$
$$\left\lceil f^1_4 + f^2_4 + f^3_4 \right\rceil \le 3,$$
$$\left\lceil f^1_1 + f^1_2 \right\rceil \le 4,$$
$$\left\lceil f^2_1 + f^2_2 \right\rceil \le 4, \quad (23)$$
$$\left\lceil f^3_1 + f^3_2 \right\rceil \le 4,$$
$$\left\lceil f^1_3 + f^1_4 \right\rceil \le 3,$$
$$\left\lceil f^2_3 + f^2_4 \right\rceil \le 3,$$
$$\left\lceil f^3_3 + f^3_4 \right\rceil \le 3.$$

(1.2) Check whether the truck departing from the logistics company can arrive within the given time and use the maximum value for travel time via

$$\max\{f_1^1, f_2^1\} \leq \left\lceil \frac{15}{130 - 45 - (20 + 15) - 45} \right\rceil,$$

$$\max\{f_1^2, f_2^2\} \leq \left\lceil \frac{15}{130 - 45 - (20 + 15) - 45} \right\rceil,$$

$$\max\{f_1^3, f_2^3\} \leq \left\lceil \frac{15}{130 - 45 - (20 + 15) - 45} \right\rceil,$$

$$\max\{f_3^1, f_4^1\} \leq \left\lceil \frac{15}{130 - 45 - (20 + 15) - 45} \right\rceil,\qquad(24)$$

$$\max\{f_3^2, f_4^2\} \leq \left\lceil \frac{15}{130 - 45 - (20 + 15) - 45} \right\rceil,$$

$$\max\{f_3^3, f_4^3\} \leq \left\lceil \frac{15}{130 - 45 - (20 + 15) - 45} \right\rceil.$$

Steps 1.1 and 1.2 generate 780 flow vectors, which are shown in the second and the third column of Table 6.

Step 2. Generate the capacity vectors $X = (x_1, x_2, \ldots, x_{10})$.

(2.1) Transform each $F$ into the capacity vector $X = (x_1, x_2, x_3, x_4)$ before the logistic company via

$$x_1 = \left\lceil f_1^1 + f_1^2 + f_1^3 \right\rceil,$$

$$x_2 = \left\lceil f_3^1 + f_3^2 + f_3^3 \right\rceil,$$

$$x_3 = \left\lceil f_2^1 + f_2^2 + f_2^3 \right\rceil,\qquad(25)$$

$$x_4 = \left\lceil f_4^1 + f_4^2 + f_4^3 \right\rceil.$$

(2.2) Transform each $F$ into the capacity vector $X = (x_5, x_6, x_7, x_8, x_9, x_{10})$ after the logistic company and use the maximal value for travel time via

$$x_5 = \left\lceil \frac{15 * \max\{f_1^1, f_2^1\}}{130 - (45 + (20 + 15) + 45)} \right\rceil,$$

$$x_6 = \left\lceil \frac{15 * \max\{f_1^2, f_2^2\}}{130 - (45 + (20 + 15) + 45)} \right\rceil,$$

$$x_7 = \left\lceil \frac{15 * \max\{f_1^3, f_2^3\}}{130 - (45 + (20 + 15) + 45)} \right\rceil,$$

$$x_8 = \left\lceil \frac{15 * \max\{f_3^1, f_4^1\}}{130 - (45 + (20 + 15) + 45)} \right\rceil,\qquad(26)$$

$$x_9 = \left\lceil \frac{15 * \max\{f_3^2, f_4^2\}}{130 - (45 + (20 + 15) + 45)} \right\rceil,$$

$$x_{10} = \left\lceil \frac{15 * \max\{f_3^3, f_4^3\}}{130 - (45 + (20 + 15) + 45)} \right\rceil.$$

This step transforms 712 flow vectors into capacity vectors $X$ in $\mathbf{\Omega}$.

(2.3) Use the method in Table 1 to eliminate other non-MCVs in $\mathbf{\Omega}$.

This step obtains 47 capacity vectors in $\mathbf{\Omega}_{\min}$, which are shown in the fourth column of Table 6.

Step 3. There are 47 MCVs that are shown in Table 6. The demand reliability $R_{(2, 2, 2)}$ in an SCCNMT according to Table 4 using the RSDP can be obtained via

$$R_{(2, 2, 2)} = \Pr\left\{ \bigcup_{i=1}^{47} \{X \mid X \geq X_i\} \right\} = 0.6321 \qquad(27)$$

**Table 6.** The results for a numerical example.

| Step 1 | Step 1.1 | Step 1.2 | Step 2 | Step 2.3 |
|---|---|---|---|---|
| Find Feasible Flow Vectors $F$ | Delete the Vectors Exceed $M_i$ | Delete the Vectors Exceed the Time Constraints | Transform $F$ to Delivery Vectors $X$ | Check if $X$ is the MCV |
| $F_1 = (0, 0, 0, 2, 0, 0, 0, 2, 0, 0, 0, 2)$ | $F_{14} = (0, 0, 0, 2, 0, 0, 1, 1, 0, 0, 0, 2)$ | $F_{14} = (0, 0, 0, 2, 0, 0, 1, 1, 0, 0, 0, 2)$ | $X_{14} = (0, 1, 0, 5, 0, 0, 0, 2, 2, 2)$ | $X_{27} = (0, 0, 3, 3, 0, 1, 2, 2, 1, 0)$ |
| $F_2 = (0, 0, 0, 2, 0, 0, 0, 2, 0, 0, 1, 1)$ | $F_{19} = (0, 0, 0, 2, 0, 0, 1, 1, 0, 0, 1, 1)$ | $F_{19} = (0, 0, 0, 2, 0, 0, 1, 1, 0, 0, 1, 1)$ | $X_{19} = (0, 2\ 0, 4, 0, 0, 0, 2, 2, 2)$ | $X_{49} = (0, 0, 3, 3, 0, 2, 1, 2, 0, 1)$ |
| $F_3 = (0, 0, 0, 2, 0, 0, 0, 2, 0, 0, 2, 0)$ | $F_{22} = (0, 0, 0, 2, 0, 0, 1, 1, 0, 2, 0, 0)$ | $F_{22} = (0, 0, 0, 2, 0, 0, 1, 1, 0, 2, 0, 0)$ | $X_{22} = (0, 1, 1, 4\ 0, 0, 0, 2, 2, 2)$ | $X_{97} = (3, 0, 0, 3, 0, 1, 2, 2, 1, 0)$ |
| $F_4 = (0, 0, 0, 2, 0, 0, 0, 2, 0, 1, 0, 1)$ | $F_{25} = (0, 0, 0, 2, 0, 0, 1, 1, 0, 1, 0, 1)$ | $F_{25} = (0, 0, 0, 2, 0, 0, 1, 1, 0, 1, 0, 1)$ | $X_{25} = (0, 1, 0, 5, 0, 0, 1, 2, 2, 1)$ | $X_{129} = (3, 0, 0, 3, 0, 2, 1, 2, 0, 1)$ |
| $F_5 = (0, 0, 0, 2, 0, 0, 0, 2, 0, 1, 1, 0)$ | $F_{31} = (0, 0, 0, 2, 0, 0, 1, 1, 1, 1, 0, 0)$ | $F_{31} = (0, 0, 0, 2, 0, 0, 1, 1, 1, 1, 0, 0)$ | $X_{31} = (1, 1, 1, 3, 0, 0, 2, 2, 2, 0)$ | $X_{277} = (0, 3, 3, 0, 0, 1, 2, 2, 1, 0)$ |
| $\vdots$ | $\vdots$ | $\vdots$ | $\vdots$ | $\vdots$ |
| $F_{998} = (2, 0, 0, 0, 2, 0, 0, 0, 1, 0, 1, 0)$ | $F_{798} = (2, 0, 0, 0, 2, 0, 0, 0, 1, 0, 1, 0)$ | $F_{798} = (2, 0, 0, 0, 2, 0, 0, 0, 1, 0, 1, 0)$ | $X_{798} = (2, 0, 3, 1, 2, 2, 1, 0, 0, 1)$ | $X_{785} = (3, 0, 0, 3, 2, 0, 1, 0, 2, 1)$ |
| $F_{999} = (2, 0, 0, 0, 2, 0, 0, 0, 1, 1, 0, 0)$ | $F_{897} = (2, 0, 0, 0, 2, 0, 0, 0, 1, 1, 0, 0)$ | $F_{897} = (2, 0, 0, 0, 2, 0, 0, 0, 1, 1, 0, 0)$ | $X_{897} = (2, 1, 3, 0, 2, 2, 1, 0, 0, 1)$ | $X_{812} = (3, 0, 0, 3, 2, 1, 0, 0, 1, 2)$ |
| $F_{1000} = (2, 0, 0, 0, 2, 0, 0, 0, 2, 0, 0, 0)$ | $F_{975} = (2, 0, 0, 0, 2, 0, 0, 0, 2, 0, 0, 0)$ | $F_{975} = (2, 0, 0, 0, 2, 0, 0, 0, 2, 0, 0, 0)$ | $X_{975} = (4, 0, 2, 0, 2, 2, 2, 0, 0, 0)$ | $X_{952} = (3, 3, 0, 0, 2, 1, 0, 0, 1, 2)$ |
| 1000 vectors | 780 vectors | 712 vectors | 712 vectors | 47 vectors |

---

**Algorithm 5:** For time reliability $R_{130}$

Step 1. Calculate the currently allocatable travel time $I^*$ via Equation (28), by referring to Table 1 for the unloading and loading time.

$$I^* = 130 - (20 + 2 \times 15) = 130 - (20 + 30) = 80. \tag{28}$$

Step 2. Find the travel time upper bound vectors.

(2.1) Find the feasible travel time vectors $V$ for each route using Equations (29) and (30),

$$
\begin{aligned}
35 \le v_1 \le 45, \\
35 \le v_2 \le 45, \\
27 \le v_3 \le 35, \\
27 \le v_4 \le 35, \\
35 \le v_5 \le 45, \\
35 \le v_6 \le 45, \\
20 \le v_7 \le 30, \\
20 \le v_8 \le 30, \\
30 \le v_9 \le 40, \\
30 \le v_{10} \le 40.
\end{aligned}
\tag{29}
$$

$$v_1 + v_5 \le 80,$$
$$v_3 + v_5 \le 80,$$
$$v_2 + v_8 \le 80,$$
$$v_4 + v_8 \le 80,$$
$$v_1 + v_6 \le 80,$$
$$v_3 + v_6 \le 80,$$
$$v_2 + v_9 \le 80,$$
$$v_4 + v_9 \le 80,$$
$$v_1 + v_7 \le 80,$$
$$v_3 + v_7 \le 80,$$
$$v_2 + v_{10} \le 80,$$
$$v_4 + v_{10} \le 80.$$

(30)

This step generates 8 travel time vectors.

(2.2) Obtain travel time vectors $V$ using step 2.1 and using the method in Table 2 to eliminate other non-travel time upper bound vectors.

In this case, the number of travel time upper bound vectors is the same as the number of travel time vectors, i.e., 7.

Step 3. There are 8 travel time upper bound vectors that are shown in Table 7. The time reliability $R_{125}$ in an SCCNMT according to Table 5 using the RSDP can be obtained via

$$R_{130} = \Pr\left\{ \bigcup_{i=1}^{8} \left\{ S \middle| S \le S_i \right\} \right\} = 0.475$$

(31)

---

**Algorithm 6:** network reliability $R_{(2, 2, 2), 130}$

$R_{(2, 2, 2), 130} = R_{(2, 2, 2)} \times R_{130} = 0.6321 \times 0.475 = 0.3001.$      (32)

---

**Table 7.** The results for a numerical example.

| Step 2.2 Travel Time Vector |
|---|
| $S_1$ = (35, 45, 35, 35, 45, 45, 30, 30, 40, 40) |
| $S_2$ = (40, 45, 35, 35, 40, 45, 30, 30, 40, 40) |
| $S_3$ = (45, 45, 35, 35, 45, 45, 30, 30, 40, 40) |
| $S_4$ = (35, 45, 35, 35, 45, 45, 30, 30, 40, 40) |
| $S_5$ = (40, 45, 35, 35, 45, 40, 30, 30, 40, 40) |
| $S_6$ = (45, 45, 35, 35, 45, 45, 30, 30, 40, 40) |
| $S_7$ = (45, 45, 35, 35, 45, 45, 30, 30, 40, 40) |
| $S_8$ = (45, 45, 35, 35, 45, 45, 30, 30, 40, 40) |
| 8 travel time upper bound vectors |

## 5. Discussion and Conclusions

Network reliability can be used as a performance indicator that indicates the SCC-NMT's ability of successfully meeting the demands of retailers and delivering within time constraints. This performance indicator can provide logistics companies with a decision-making basis for determining transportation routes and vehicle configurations, and can also be used for sensitivity analysis to identify significant transportation routes. Although we consider travel time during transportation, only travel data on highways are available from public data. To be more comprehensive, we need travel time data on both highways and surface roads for more realistic and accurate performance indicators.

In this article, we introduce a SCC model, which models the entire product chain and calculates network reliability using stochastic flow. Additionally, because travel time is a crucial factor in a cold chain network, we need to consider it as an additional factor by calculating time reliability and multiplying it with the multi-state flow to estimate network reliability. This way, we can evaluate network reliability not only by including multi-state flow but also considering the multi-state travel time. Currently, this research only considers a single product in a cold chain network, but in the future, it can be extended to study the inventory service and multiple commodities.

**Author Contributions:** The authors contributed equally to the work. All authors have read and agreed to the published version of the manuscript.

**Funding:** National Science and Technology Council Taiwan: 109-2221-E-009-067-MY3 and 111-2221-E-167-010.

**Data Availability Statement:** The data used can be obtained from the references cited.

**Conflicts of Interest:** The authors declare no conflict of interest.

## Notations and Assumptions

| | |
|---|---|
| **N** | set of nodes consisting of suppliers, the logistic companies and retail stores. |
| $s$ | number of arcs from the supplier to the logistics company. |
| $z$ | number of arcs in the network. |
| $a_q$ | $q$th arc in the network, $q = 1, 2, \ldots, s, s+1, \ldots, z$. |
| **A** | $\{a_q \mid q = 1, 2, \ldots, z\}$: set of arcs. |
| $M_q$ | maximal capacity for arc $a_q$, $q = 1, 2, \ldots, z$. |
| $M$ | $(M_1, M_2, \ldots, M_z)$: maximal arc-capacity vector. |
| $G$ | $(\mathbf{N}, \mathbf{A}, M)$: an SCCNMT. |
| $v_q$ | travel time of arc $a_q$, $q = 1, 2, \ldots, z$. |
| $x_q$ | current capacity of arc $a_q$, $q = 1, 2, \ldots, z$. |
| $X$ | $(x_1, x_2, \ldots, x_z)$: current capacity vector. |
| $U$ | average unloading time required for a truck at the logistic company and retail store. |
| $L$ | average loading time required for a truck at the logistic company and retail store. |
| $p$ | number of retail stores. |
| $d^k$ | demand of the retail store $k$, $k = 1, 2, \ldots, p$. |
| $D$ | $(d^k \mid k = 1, 2, \ldots, p)$: demand vector for each retail store. |
| $w$ | number of delivery paths from supplies to retail stores. |
| $P_o^k$ | the $o$th delivery path for retail store $k$, $o = 1, 2, \ldots, w$; $k = 1, 2, \ldots, p$. |
| $f$ | flow on $P_o^k$, $o = 1, 2, \ldots, w$; $k = 1, 2, \ldots, p$. |
| $F$ | $(f_1^1, f_2^1, \ldots, f_w^1, f_1^2, f_2^2, \ldots, f_w^2, f_1^p, f_2^p, \ldots f_w^p)$: flow vector. |
| $I$ | the time threshold. |
| $I^*$ | the currently available travel time threshold. |
| $R_D$ | demand reliability that an SCCNMT can satisfy the demand with maximal travel time. |
| $\Omega$ | set of capacity vectors satisfying the demand and time constraints. |
| $\Omega_{\min}$ | set of minimal capacity vectors $X$. |
| $V$ | travel time vector under the feasible time threshold. |
| $v_q^{\min}$ | minimum travel time of arc $a_q$, $q = 1, 2, \ldots, z$. |
| $v_q^{\max}$ | maximum travel time of arc $a_q$, $q = 1, 2, \ldots, z$. |
| **S** | set of travel time upper bound vector. |
| $R_T$ | time reliability that an SCCNMT can deliver within the time threshold. |
| $R_{D,T}$ | network reliability to satisfy the demand within the travel time threshold. |

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
