# Peer review of "Estimation of the Network Reliability for a Stochastic Cold Chain Network with Multi-State Travel Time"

_applsci, doi:10.3390/app13137897_

Round 1
Reviewer 1 Report
Very well written. Useful pseudo code.
Author Response
Thanks for your appreciation!
Reviewer 2 Report
In this paper, the authors proposed an algorithm that calculate the reliability of for a stochastic cold chain network as the multiplication of demand reliability and time reliability, where both multi-state flow and multi-flow travel time are considered. The paper is organized well and is ready to publish after some minor editorial work.
The paper needs some minor editorial work. The authors should check thoroughly. I only point out a few:
1. Page 6, Line 154: "satisfy" should be "satisfying";
2. Page 7, Line 199, replace "There is" with "We use";
3. Page 7, Line 203, replace "lists" with "list".
Author Response
According to your comment, we have checked carefully to remove all typos and grammar mistakes from the manuscript. Some of the modifications are listed below:
- Page 6, Line 154: “satisfy” should be “satisfying”;
- Page 7, Line 199, replace “There is” with “We use”;
- Page 7, Line 203, replace “lists” with “list”.
- Page 1, Line 27, replace “all companies” with “companies”.
- Page 2, “the aq” should be “arc aq”.
- Page 4, Line 121, “enumerating all X in Ω” should be “enumerating all X in the set Ω”
- Page 4, Line 124, “MDVs” should be “MCVs”.
- Page 5, Line 148, “upper bound vector” should be “upper bound vectors”.
- Page 9, Line 217, “These step generates” should be “This step generates”.
- Page 10, Line 224, “The steps 1.1 and 1.2” should be “The step 1.1 and 1.2”.
Finally, thank you very much for your comments that help improve the quality of the paper.
Reviewer 3 Report
This paper presents an SCC model that models the entire supply chain and calculates the network's reliability using the stochastic flow. Since travel time is a critical factor in a cold chain network, it must be considered an additional factor by calculating the temporal reliability and multiplying it by the multi-state flow to estimate its reliability. In this way, the network's reliability can be evaluated by considering the multi-state flow and the multi-state travel time. Currently, this study finds only a single product in a cold chain network, but in the future, it can be extended to study inventory service and multiple commodities. The article is well-written, and the results are interesting. I recommend the article for publication in Applied Sciences.
Author Response
Thanks for your comments on our submission. We will extend this model to study inventory service and multiple commodities in near future!
Reviewer 4 Report
The article is interesting and the researched problem has scientific potential. However, some problems need to be solved:
1. The paper needs a Literature review section. Literature review should include more recent sources (2020-2023) and be enriched with relevant references.
2. The authors must briefly present the steps of the research (possibly in a figure).
3. In my opinion, a discussion section that includes theoretical and managerial implications, research limitations, and future research directions would be helpful.
The article has scientific value and can be published after carefully reviewing the reported issues.
Author Response
1. According to your comment, we have added some recent research in section 1.2 and discussed their relevance to our research.
“Previous research on stochastic supply chain networks has explored various aspects of network reliability. Huang [8] studied the different states of available transportation on different roads and proposed a network performance evaluation algorithm for inventory issues. Lin et al. [7] investigated the use of different transportation modes in the network and considered the characteristic of goods being damaged. Niu et al. [12] further examined the cost concept by incorporating transportation and damage costs into the study and separately discussing the damage rates on different routes. Lin et al. [9] specifically discussed perishable goods, calculated the flow rate that meets the damage rate and ensured that the product can meet demand under the possibility of damage. As stochastic networks have been studied by previous studies with only a single multi-state factor, this study attempts to extend the research by evaluating the network reliability under two multi-state factors.”
2. Thanks for your comment. To make the entire algorithm clearer and easier to understand, we have represented it as a diagram shown in Figure 2 on page 8.
3.Thank you for your comment. The authors have added some discussions including managerial implications, research limitations, and future research directions to the conclusion section.
“Network reliability can be used as a performance indicator that indicates the SCCNMT's ability of successfully meeting the demands of retailers and delivering within time constraints. This performance indicator can provide logistics companies with a decision-making basis for determining transportation routes and vehicle configurations and can also be used for sensitivity analysis to identify significant transportation routes. Although we consider travel time during transportation, only travel data on highways is available from public data. To be more comprehensive, we need travel time data on both highways and surface roads for more realistic and accurate performance indicators.”
Finally, thank you very much for your comments that help improve the quality of the paper.